# Spongy-Network-like Polyaniline Thin Films as Electrodes for a Supercapacitor

P. M. Kharade [1],*, J. V. Thombare [2], S. S. Dhasade [2], S. S. Deokar [3], D. J. Salunkhe [4], Mohaseen S. Tamboli [5] and Santosh S. Patil [6],*

1 Department of Physics, Shankarrao Mohite Mahavidyalaya College, Akluj 413101, India
2 Department of Physics, Vidnyan Mahavidyalaya College, Sangola 413307, India
3 Department of Chemistry, Shankarrao Mohite Mahavidyalaya College, Akluj 413101, India
4 Nanocomposite Research Laboratory, K.B.P. Mahavidyalaya College, Pandharpur 413303, India
5 Korea Institute of Energy Technology (KENTECH), 200 Hyeokshin-ro, Naju 58330, Korea
6 Department of Chemistry and Chemical Engineering, Inha University, 100 Inha-ro, Michuhol-gu, Incheon 22212, Korea
* Correspondence: pravink150@gmail.com (P.M.K.); santoshkumar.patil.19@gmail.com (S.S.P.)

**Abstract:** An easy and cost-effective route is demonstrated to grow spongy-network-like polyaniline (SpN-PANI) thin films on stainless steel (SS) by galvanostatic electrodeposition. Through X-ray diffraction (XRD), Fourier transform infrared (FTIR) spectroscopy, and scanning electron microscopy (SEM) characterizations, the physicochemical properties of the SpN-PANI thin films were fine-tuned for supercapacitor application. The hydrophilic nature of SpN-PANI thin films was examined by using contact angle measurements. Next, the capacitive behavior of the SpN-PANI thin film was assessed by using cyclic voltammetry (CV), galvanostatic charging-discharging (GCD), and electrochemical impedance spectroscopy (EIS). Specifically, these results show that SpN-PANI thin films can exhibit a maximum specific capacitance of 580 F.g$^{-1}$ at a scan rate of 5 mV.s$^{-1}$ in a 0.5 M Na$_2$SO$_4$ electrolyte solution, as well as superior cycling stability (84% capacity retention after 1000 cycles). Thus, the strategy presented here can be applicable to produce a SpN-PANI-based thin film which has prospects as an active electrode material for supercapacitor devices.

**Keywords:** electrodeposition; supercapacitor; PANI; CV; GCD; EIS

## 1. Introduction

Over the last few years, naturally conducting polymers have attracted significant scientific and technological attention due to their potential applications. Among all the conducting polymer families, PANI is the most promising material due to easy synthesis, low price, high electronic conductivity, distinctive doping/dedoping properties, good redox reversibility, environmental stability, mechanical flexibility, etc. [1]. PANI can be used in various potential applications such as sensors [2], light emitting diodes [3], electrochemical supercapacitors [4,5], nonlinear optical devices [6], etc.

PANI thin films have been synthesized by different chemical and physical methods. Interestingly, different synthesis methods give distinct surface morphology of PANI thin films, such as nanocapsules [7], nanofibers [8], nanorods [9], nanowires [10], nanospheres [11], and nanograins [12]. The electrochemical deposition (electro polymerization) method is one of the best methods for the deposition of conducting polymers due to it being a simple and reasonable method. It enables control over the thickness and morphology of electrodes by altering the deposition condition [13]. According to the applied voltage or current, it can be classified into three main categories: potentiostatic, galvanostatic, and potentiodynamic electrodeposition methods [14]. Previously, Jamadade et al. [15] reported a maximum specific capacitance of 258 F.g$^{-1}$ at a scan rate of 10 mV.s$^{-1}$ for electrosynthesized emeraldine (E) forms of PANI onto stainless steel (304) substrates. Similarly, Yang et al. [7] carried

out the synthesis of hollow polyaniline nanocapsules using the interfacial polymerization method and applied supercapacitor application, which exhibits a high specific capacitance of 502 $F.g^{-1}$ at a constant current density of 5 $mA.cm^{-2}$. Inamdar et al. [16] synthesized PANI thin films using the electrodeposition (ED) method from a mixed solution of 0.2 M aniline and 0.2 M $H_2SO_4$, which showed the highest specific capacitance of 473 $F.g^{-1}$ at a constant current density of 1 $mA.cm^{-2}$.

In this work, a galvanostatic ED strategy is investigated to synthesize SpN-PANI thin films on a stainless steel (SS) substrate. SS substrates are usually used for applications in supercapacitors due to their good conductivity, high mechanical strength, flexibility, light weight, and low cost. SpN-PANI thin films were characterized by XRD, FTIR spectroscopy, SEM, and contact angle measurement. Further, we investigated the electrochemical supercapacitive performance of synthesized SpN-PANI thin films by using CV, GCD, and EIS analysis. This shows that SpN-PANI thin films are electrodes with good electronic conductivity, higher values of specific capacitance, coulombic efficiency, and reversibility. In particular, the electrochemical performance of the SpN-PANI thin film electrode is found to be strongly dependent on its surface morphology. This unique nanostructure and porous morphology of the PANI electrode can not only offer highly active surface sites and high depth penetration of ions from electrode to electrolyte but also likely promotes rapid electron transport, which is of great interest in electrochemical supercapacitor application [7].

## 2. Materials and Methods

### 2.1. Synthesis of SpN-PANI Thin Film

In earlier work [4], we used 0.2 M concentrations of both aniline and $H_2SO_4$. In the present study, we used 0.1 M concentrations of both aniline and $H_2SO_4$, which enables proper variations in the structure of the polyaniline chain.

Before electrodeposition of the SpN-PANI thin film, the SS substrate was polished with emery polish paper rough to finish, and further followed by rinsing with water and cleaning with acetone. For synthesis of the SpN-PANI thin film, 0.1 M aqueous aniline solution was added to a 0.1 M aqueous $H_2SO_4$ solution. The mixed solution was stirred a few times to prepare a homogeneous solution for electrodeposition. The SpN-PANI thin film was deposited onto SS substrates via the galvanostatic mode of ED at a constant current density of 5 mA for 600 s at room temperature. Figure 1 illustrates the schematic representation of the electrodeposition process for the synthesis of the polyaniline thin film. The as-deposited SpN-PANI thin film is greenish blue in color, indicative of the formation of emeraldine salt of the PANI thin film.

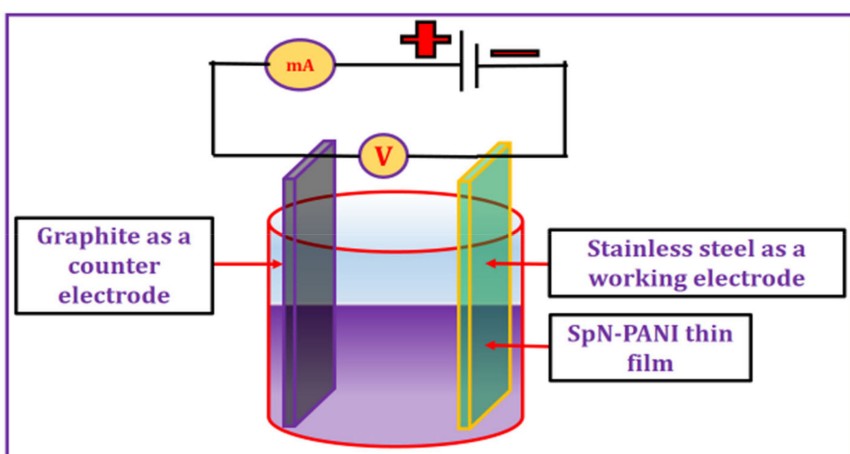

**Figure 1.** Schematic representation of electrodeposition set for the growth of polyaniline thin film.

## 2.2. Characterization Techniques

The structural study of the SpN-PANI thin film was studied by X-ray diffraction (XRD) techniques and Fourier transform infrared (FTIR) spectroscopy using Bruker axes D8 Advance Model with copper radiation ($K_\alpha$ of $\lambda = 1.54$ A°) in the 2θ range between 20° to 80° and using a Perkin Elmer, FTIR Spectrum one unit. The surface morphology of the SpN-PANI thin film was studied using SEM, JEOL JSM 6390. The surface wettability test of the SpN-PANI thin film was carried out by measuring the contact angle with a water droplet using a Rame-Hart contact angle meter. The supercapacitive study of the SpN-PANI thin film was carried out by cyclic voltammetry, charge-discharge, and electrochemical impedance using an electrochemical workstation (CHI 660C). An electrochemical cell was comprised three electrode systems, the SpN-PANI thin film as a working electrode, graphite as a counter electrode, and saturated calomel electrode (SCE) as a reference electrode. The aqueous 0.5 M $Na_2SO_4$ was used as the electrolyte in the electrochemical cell.

## 3. Results

### 3.1. Structural and Surface Morphology Study

Figure 2a shows the XRD pattern of SS and SpN-PANI thin films. The XRD spectra of the SpN-PANI thin film indicates the amorphous nature of polymer films. In addition, the peaks indicated by an asterisk are SS substrates only. This amorphous nature of the SpN-PANI thin film can be useful for the supercapacitive study, as reported in previous studies by Dhawale et al. [5]. As shown in Figure 2b, the FTIR spectra of the SpN-PANI thin film was acquired, which indicates the formation of the conducting polymer films. According to the FTIR spectra, the SpN-PANI thin film shows characteristics peaks at 3209, 3049, 1584, 1498, 1309, 1041, 861, and 753 cm$^{-1}$. The peak at 3209 cm$^{-1}$ corresponds to N-H stretching modes, which occur due to protonation of the SpN-PANI thin film [17]. The transmission peaks at 1584 and 1498 correspond to the C=C quinoid and benzenoid vibration. The peak at 1309 is attributed to the C-N imine stretching vibration. In addition, the peak at 1041 corresponds to C–N stretching of the secondary aromatic amine and the C–H aromatic in-plane bending [5]. The peak at 861 and 753 can be attributed due to the C=H out-of-plane deformation in the 1,4-disubstituted benzene ring [18]. Thus, emeraldine salt of the SpN-PANI thin film was confirmed from the FTIR study.

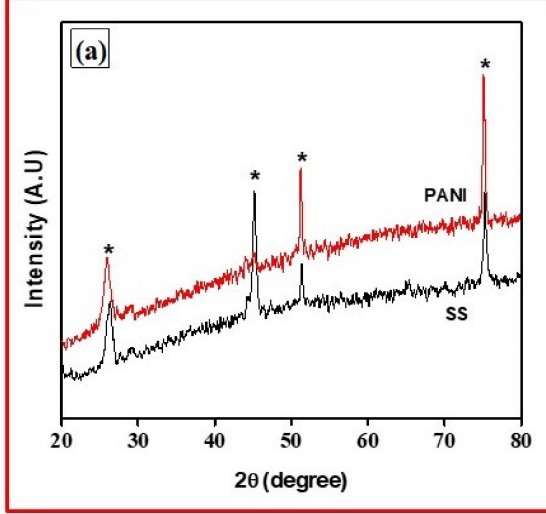 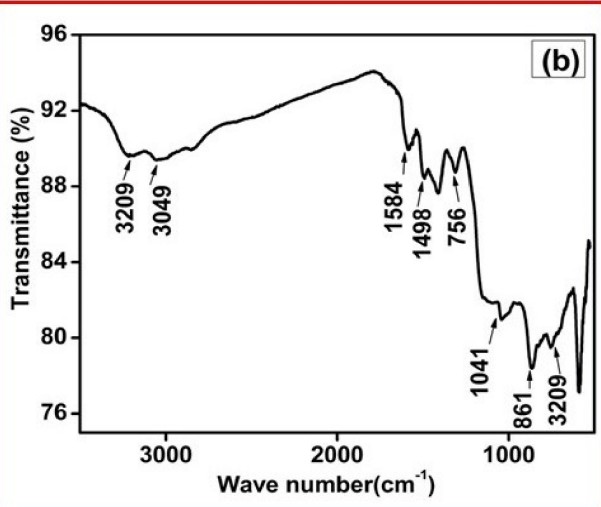

**Figure 2.** (**a**) XRD pattern of SS and SpN-PANI thin film, (**b**) FTIR spectrum of SpN-PANI thin film.

The surface morphology features of the SpN-PANI thin film were carried out by using SEM at two different magnifications: 25 kX and 50 kX (Figure 3). It is noteworthy that the electro polymerization of aniline mostly involved two stages. Firstly, a compact granular structure of PANI is formed on the SS electrode in the primary stage. Secondly, PANI further

grows and then forms a uniform and loosely bound morphology in the advanced stage, which seems to be a spongy-network-like structure [17]. As can be seen from Figure 3a at lower magnification, the surface of the SpN-PANI thin film is fully covered with a spongy network. The unique spongy-network-like structure with micro-pores is uniformly coated onto an SS substrate. These micro-pores diameters were observed to be in the size range of 300 nm to 500 nm, which is clearly shown in Figure 3b. This spongy-like structure is expected to provide a larger surface area for the electrochemical reaction due to micro-pores which provide a fast flow of charge carriers. These are key requirements in a supercapacitor, which offers a high surface area; it also diminishes the diffusion resistance of the electrolyte into the electrode, thereby enhancing the supercapacitive performance [19]. Surface wettability of the SpN-PANI thin film was examined by performing contact angle measurements, as displayed in the inset of Figure 3b. The value of the contact angle was observed to be 30°, indicating that the SpN-PANI thin film is hydrophilic in nature. The hydrophilic nature of the electrode is helpful for improving the supercapacitive performance. Previously, Dhawale et al. [5] reported that the water contact angle for polyaniline thin film is 15° for supercapacitor application. In the present work, the value of the contact angle is relatively high, and this may be due to the formation of the spongy-network-like structure of the polyaniline thin film.

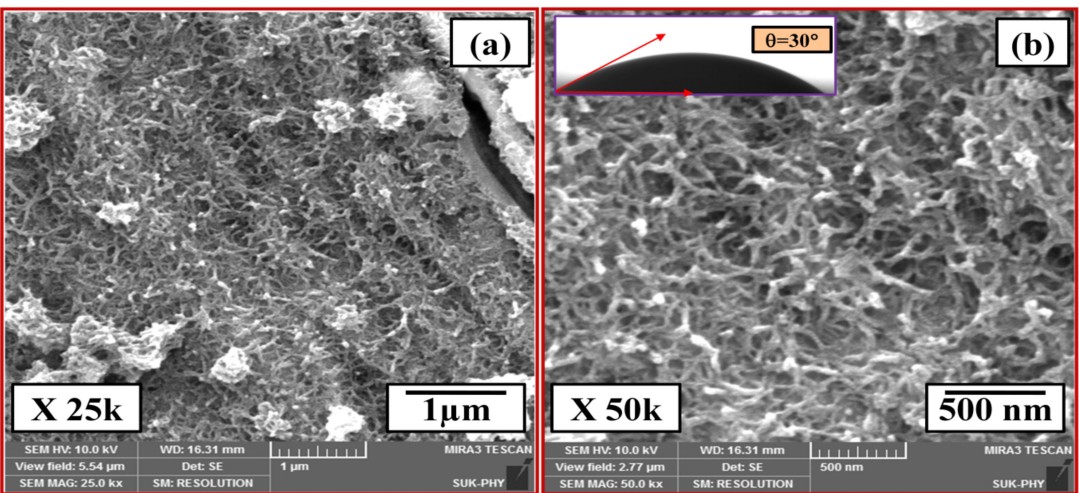

**Figure 3.** SEM images of SpN-PANI thin film at (**a**) 25 kX and (**b**) 50 kX magnifications, respectively. Inset of (**b**): contact angle of SpN-PANI thin film with water droplet.

### 3.2. Cyclic Voltammetry (CV) Study

Figure 4A shows typical CV curves of the SpN-PANI thin film at two different scan rates, 5 and 50 mV.s$^{-1}$, within the potential window of +1 V to −1 V, respectively. It was observed that as the scan rate increased, the current under the curve increased, i.e., current density is directly proportional to scan rate. This also indicates the ideal faradaic supercapacitive behavior of SpN-PANI thin film.

The volumetric capacitance was calculated by using the formula:

$$C_v = \frac{\int Idt}{V * \Delta V} \tag{1}$$

where V is volume of the electrode in cm$^3$ and $\Delta$V is the voltage window in volts.

The volumetric capacitance of the SpN-PANI thin film was found to be 10.8 F.cm$^3$.

The specific capacitance of the SpN-PANI thin film was calculated from CV curves using the formula reported in earlier literature [4].

$$C = \frac{\int Idt}{dv/dt} \tag{2}$$

where $\int$ Idt is the area under curve of CV and dv/dt is the voltage scanning rate in mV/s.

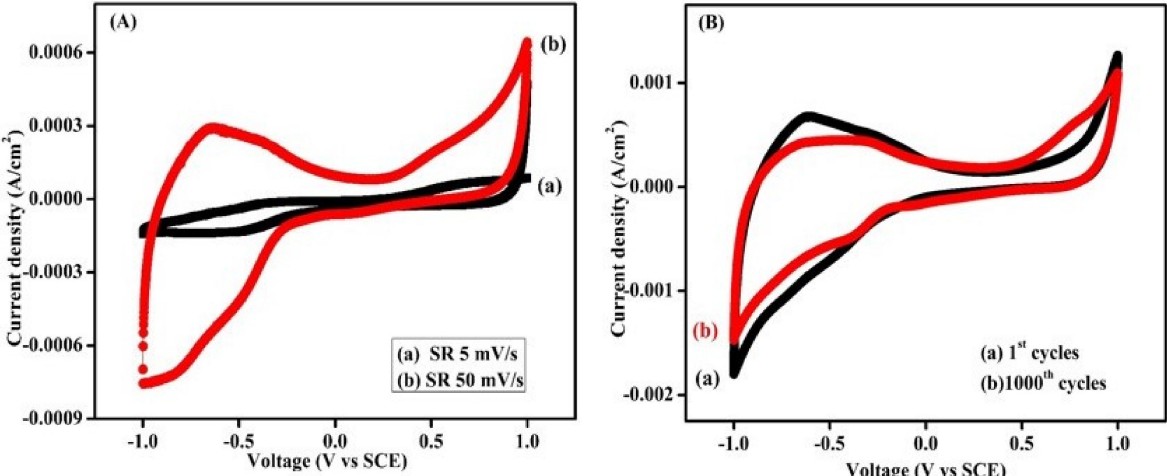

**Figure 4.** (**A**) CV study of SpN-PANI thin film at scan rate of 5 and 50 mV/s and (**B**) stability of SpN-PANI thin film at 1st and 1000th cycles.

Our SpN-PANI thin film showed a maximum specific capacitance of 580 F.g$^{-1}$ at a scan rate of 5 mV/s in a 0.5 M Na$_2$SO$_4$ electrolyte solution. It was also found that specific capacitance decreases with an increase in scan rate. This is because at a higher scan rate, most of the inner active species of the spongy network do not take part in the reaction, and only the exterior part is actively participating. Table 1 represents a comparison of specific capacitance with previously reported work by keeping a scan rate of 5 mV/s vs. SCE. In the present study, the higher value of specific capacitance may be ascribed to the unique spongy-network-like nanostructure, which may endow the generation of more active sites and hydrophilic behavior of the SpN-PANI thin film.

**Table 1.** Comparison of specific capacitance between our work and previously reported work to synthesize the polyaniline electrodes.

| Material | Conc. | Specific Capacitance (F.g$^{-1}$) | Electrolyte | Method of Synthesis | Reference |
|---|---|---|---|---|---|
| Polyaniline | 0.2 M Aniline + 0.2 M H$_2$SO$_4$ | 473 | 0.5 M H$_2$SO$_4$ | Electrodeposition | [16] |
| Polyaniline nanograins | 0.2 M Aniline + 0.2 M APS + 0.2 M H$_2$SO$_4$ | 503 | 1.0 M H$_2$SO$_4$ | Template-free and seedless method | [12] |
| Polyaniline nanofibers | 0.1 M Aniline + 0.1 M H$_2$SO$_4$ | 508.7 | 1.0 M H$_2$SO$_4$ | Electrodeposition | [20] |
| Polyaniline nanotube | 0.2 mL Aniline + 1 M HCl + 0.1 g of K$_2$Cr$_2$O$_7$ | 510 | 1.0 M H$_2$SO$_4$ | Chemical Polymerization | [21] |
| Polyaniline nanocapsules | 0.96 g Aniline + 0.57 APS | 502 | 1.0 M H$_2$SO$_4$ | Interfacial Polymerization | [7] |
| SpN-polyaniline | 0.1 M Aniline + 0.1 M H$_2$SO$_4$ | 580 | 0.5 M Na$_2$SO$_4$ | Galvanostatic Electrodeposition | This work |

*3.3. Stability Study*

Together with specific capacitance, the cyclability is also a critical parameter in supercapacitor device application. Therefore, the long-term cycling performance of the electrode was performed to test the supercapacitor performance. Figure 4B shows the cyclic stability of the SpN-PANI thin film measured at a scan rate of 100 mV.s$^{-1}$ for the 1st and 1000th

cycles in a 0.5 M Na$_2$SO$_4$ electrolyte solution. As seen in Figure 4B, the slight decrease in specific capacitance with the increasing cycle number of the SpN-PANI thin film was ascribed to the loss of active sites of the spongy network affected by dissolution and/or detachment during the cycling process in the electrolyte solution [5]. The synthesized SpN-PANI thin film shows 84% cycling stability at the 1000th cycle, which shows the potential utility of this material for energy storage device applications.

### 3.4. Galvanostatic Charge-Discharge (GCD) Study

Figure 5A shows a typical galvanostatic charge-discharge curve for an SpN-PANI thin film electrode at a constant current density of 1 mA.cm$^{-2}$ in a 0.5 M Na$_2$SO$_4$ electrolyte solution. The nature of this GCD curve is not ideal and triangular. At the beginning, the discharge curve shows a small potential drop because of internal resistance between the SpN-PANI thin film and the electrolyte.

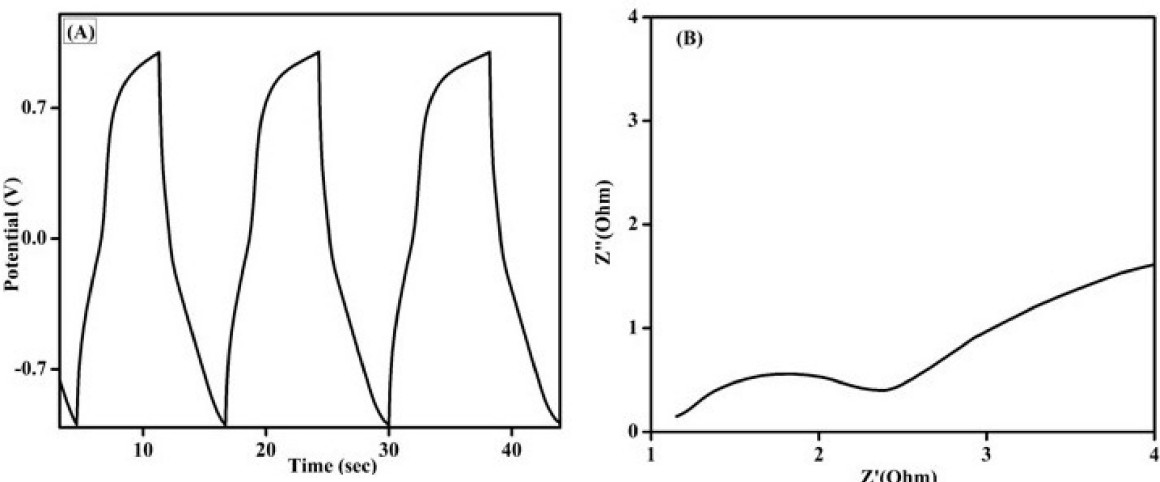

**Figure 5.** (**A**) GCD study of SpN-PANI thin film and (**B**) EIS study of SpN-PANI thin film.

The electrochemical supercapacitor parameters such as coulombic efficiency, specific energy, and specific power associated with the SpN-PANI thin film was calculated by using following formula [22]:

$$\text{Coulombic Efficiency} = \frac{T_d}{T_c} \times 100 \tag{3}$$

$$\text{Specific power} = \frac{V \times I_d}{W} \tag{4}$$

$$\text{Specific energy} = \frac{V \times I_d \times T_d}{W} \tag{5}$$

where $T_d$ is discharging time in sec and $T_c$ is charging time in sec, V is voltage window volt, $I_d$ is discharging current in A, and W is the weight of active material in gm.

The SpN-PANI thin film electrode showed a coulombic efficiency of 95.22%, and specific power and specific energy were found to be 5.35 kW.kg$^{-1}$ and 3.40 Wh.kg$^{-1}$, respectively.

### 3.5. Electrochemical Impedance Spectroscopy (EIS) Study

The electrochemical impedance spectroscopy (EIS) technique was carried out to investigate reaction kinetics and charge transfer properties. Figure 5B shows a Nyquist plot of the SpN-PANI thin film within a frequency range of 10 Hz to 1 MHz in a 0.5 M Na$_2$SO$_4$ electrolyte solution. The Nyquist plot of the SpN-PANI thin film comprises a semicircle in the high frequency region and a vertical line in the low frequency region. The semicircle

in the high frequency region corresponds to a value of series resistance ($R_s$) composed of the accumulation of the intrinsic resistance of the active material, solution resistance, and contact resistance between the SpN-PANI thin film/electrolyte interface. The diameter of the semicircle provides the value of charge transfer resistance ($R_{ct}$). The straight line in the low frequency region indicates the ion diffusion process of the SpN-PANI thin film [23].

The values of $R_s$ and $R_{ct}$ of the SpN-PANI thin film were observed to be 1.07 Ω and 2.35 Ω, respectively. Small values of $R_s$ and $R_{ct}$ of the SpN-PANI thin film reflect good electronic conductivity and the fast transfer of ions from the electrolyte to the SpN-PANI thin film, which results in good electrochemical supercapacitor performance of the SpN-PANI thin film.

## 4. Conclusions

In summary, we investigated the feasibility of polyaniline thin film synthesis with a spongy-network-like structure with an electrodeposition strategy. Based on physicochemical analysis, we discovered that SpN-PANI thin films have a porous morphology and a greater number of active sites, which can greatly promote electrochemical reactions and capacitive performance. XRD, FTIR, and EIS spectroscopy together give the evidence of formation of conducting polyaniline thin film and the fast movement of electronic and ionic conductivity across the electrode–electrolyte interface. Importantly, the SpN-PANI thin film electrode exhibited a wide potential window of 2.0 V, enabling a significantly higher specific capacitance of 580 F.g$^{-1}$ at a scan rate of 5 mV.s$^{-1}$ and excellent cycling stability (84% retention after 1000 cycles), which has great potential to be used in futuristic supercapacitor devices.

**Author Contributions:** Conceptualization, P.M.K.; formal analysis, J.V.T. and S.S.D. (S. S. Dhasade); investigation, P.M.K. and S.S.D. (S. S. Deokar); data curation, M.S.T. and S.S.P.; writing—original draft preparation, P.M.K. and D.J.S.; writing—review and editing, P.M.K., J.V.T. and S.S.P.; supervision, D.J.S.; project administration, P.M.K. All authors have read and agreed to the published version of the manuscript.

**Funding:** This research received no external funding.

**Institutional Review Board Statement:** Not applicable.

**Informed Consent Statement:** Not applicable.

**Data Availability Statement:** Not applicable.

**Acknowledgments:** P.M.K. is grateful to D. S. Bagade, S. M. M. Akluj, for his consistent research support.

**Conflicts of Interest:** There is no conflict of interest.

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
