# Peer review of "Spongy-Network-like Polyaniline Thin Films as Electrodes for a Supercapacitor"

_2673-8023, doi:10.3390/micro2030035_

Round 1

Reviewer 1 Report

In the present study, the authors grew network like Polyanilne (SpN-PANI) thin films on stainless steel (SS) by galvanostatic electrodeposition for applications in supercapacitor. Systematic characeterizations are performed and the results are interesting. I would recommend the acceptance of current manuscript after addressing the following issues:

1) The volumetric capacitance should be calculated and discussed.

2) Most commercial supercapacitors use 6M KOH as the electrolyte. The electrochemical activity should be also measured in 6M KOH to show their potential use for practical application.

3) Is it possible to provide the BET result of the PANI film?

4) The role of stainless steel should be studied. The author are expected to use other type of metal substrate (e.g., spongy nickel) or carbon paper to prepare the PANI film and test the performance.

Author Response

Reviewer #1 comments:

In the present study, the authors grew network like Polyanilne (SpN-PANI) thin films on stainless steel (SS) by galvanostatic electrodeposition for applications in supercapacitor. Systematic characeterizations are performed and the results are interesting. I would recommend the acceptance of current manuscript after addressing the following issues:

Comment-1:  The volumetric capacitance should be calculated and discussed.

Response: Thank you for helpful comments. The volumetric capacitance of SpN-PANI thin film was calculated and discussed in cyclic voltammetric section in revised manuscript.

Comment-2: Most commercial supercapacitors use 6M KOH as the electrolyte. The electrochemical activity should be also measured in 6M KOH to show their potential use for practical application.

Response: Thank you for reviewers helpful suggestion. We were interested in ordinary electrolytes instead of commercial electrolytes which have been reported previously for supercapacitive study. In future work we will consider studying the electrodes for various electrolytes with different concentrations like 6 M KOH.

Comment-3: Is it possible to provide the BET result of the PANI film?

Response: Thank you for your helpful comment. We agree that BET results are useful to analyze PANI film, but unfortunately, we cannot provide BET characterization because of lack of facility.

Comment-4: The role of stainless steel should be studied. The author are expected to use other type of metal substrate (e.g., spongy nickel) or carbon paper to prepare the PANI film and test the performance.

Response: Thank you for reviewers helpful suggestion. Stainless steel (SS) substrate plays an important role in supercapacitors due to their good conductivity, high mechanical strength, flexibility and low cost. In future work, we are trying to study other type of metal substrate (e.g., spongy nickel) or carbon paper to prepare the PANI film and test the performance.

Reviewer 2 Report

1. The novelty of this work can be hardly found. PANI has been studied as supercapacitor electrodes in many previous studies, and the synthesis method used in this paper was similar to the previous work (reference 4).

2. Why was a SS substrate chosen? What is the area and thickness of the SS? Authors should describe the SS substrate more.

3. In XRD pattern, authors explained these peaks were only from SS. It so, XRD pattern of SS only should be indicated for comparison.

4. Cross-sectional SEM images of SpN-PANI would be necessary to see if the surface of SS was fully covered by PANI.

5. Authors should explain how spongy network like structure was formed by this synthesis method.

6. What is the current density of GCD test?

7. It would be better to add "spongy" in the first sentence of Abstract.

Line 15, Page 1) An easy and cost-effective route is demonstrated to grow spongy network like Polyanilne (SpN-PANI)

8. It doesn’t seem necessary to use capital letters:

Line 15, Page 1) Polyanilne to polyanilne

Line 17, Page 1) Fourier Transform Infrared to Fourier transform infrared

 Scanning Electron Microscopy (SEM) to scanning electron microscopy (SEM)

9. There are several grammatical errors and typos. For example,

Line 19, Page 1) supercapacitor application. Contact angle measurements suggests surface wettability of synthesized

Line 16, Page 16) ofof to of

In addition, several missing prepositions were shown.

10. Is it necessary to use "." between two units? F g-1 and mV s seem more natural. 

Author Response

Reviewer #2 comments:

Comment-1: The novelty of this work can be hardly found. PANI has been studied as supercapacitor electrodes in many previous studies, and the synthesis method used in this paper was similar to the previous work (reference 4).

Response: Thank you reviewer for your valuable comment. We agree with your comment. But synthesis parameters can change the properties of polymers. In earlier work (reference 4), we have used 0.2 M concentrations of both aniline and H2SO4. In the present study we have taken 0.1 M concentrations of both aniline and H2SO4. So, we get proper variations in the structure of polyaniline chain.

Comment-2: Why was a SS substrate chosen? What is the area and thickness of the SS? Authors should describe the SS substrate more.

Response: Thank you reviewer for helpful comments, stainless steel (SS) substrate is commonly used for application in supercapacitors due to their good conductivity, mechanical strength, flexibility, and low cost. The area of SS substrate is 5 cm2 and thickness is 0.2 mm.

Comment-3: In XRD pattern, authors explained these peaks were only from SS. It so, XRD pattern of SS only should be indicated for comparison.

Response: Thank you reviewer for helpful suggestion. The XRD pattern of SS is also added for comparison (please see Fig. 2a).

Comment-4: Cross-sectional SEM images of SpN-PANI would be necessary to see if the surface of SS was fully covered by PANI.

Response: Thank you for your valuable comments. Cross-sectional SEM image is important to study surface morphology of SpN-PANI, but unfortunately, we have not taken cross-sectional SEM images and it is not possible for us now because of lack of facility.

Comment-5: Authors should explain how spongy network like structure was formed by this synthesis method.

Response: Thank you reviewer for your valuable comments. It is noteworthy that, the electro polymerization of aniline mostly involved into two stages. Firstly, a compact granular structure of PANI is formed on the SS electrode in the primary stage. Secondly, PANI further grows and then forms a uniform and loosely bound morphology in the advanced stage which seems to be spongy network like structure.

Comment-6: What is the current density of GCD test?

Response: Thank you reviewer for your valuable comments,

The current density of GCD test is 1 mA.cm-2

Comment-7: It would be better to add "spongy" in the first sentence of Abstract.

Line 15, Page 1) An easy and cost-effective route is demonstrated to grow spongy network like Polyanilne (SpN-PANI)

Response: Thank you reviewer for reviewers helpful suggestion. Accordingly, the first sentence of abstract has been modified.

Comment-8: It doesn’t seem necessary to use capital letters:

Line 15, Page 1) Polyanilne to polyanilne

Line 17, Page 1) Fourier Transform Infrared to Fourier transform infrared

 Scanning Electron Microscopy (SEM) to scanning electron microscopy (SEM)

Response: Thank you reviewer for reviewers’ helpful suggestion. We have revised the line 15 and 17 of page 1 polyaniline, Fourier transform infrared and scanning electron microscopy (SEM).

Comment-9: There are several grammatical errors and typos. For example, Line 19, Page 1) supercapacitor application. Contact angle measurements suggests surface wettability of synthesized

Line 16, Page 16) ofof to of In addition, several missing prepositions were shown.

Response: Thank you for reviewers helpful suggestion. In the modified version, we have revised line 19 of page 1 as, Hydrophilic nature of SpN-PANI thin was examined by using contact angle measurements study and line 16, page 16 of.

Comment-10: Is it necessary to use "." between two units? F g-1 and mV s seem more natural. 

Response: Thank you reviewers’ valuable comment. Yes, it is necessary to use “.” between two units.

Round 2

Reviewer 1 Report

The authors did not response to my concerns seriously and appropriately. I would like to give them a second chance to revise the manuscript carefully. 

Author Response

The following are the responses to the reviewers comments for the manuscript “Spongy Network like Polyaniline Thin Films as an Electrodes for Supercapacitor” Ref: micro-1831345.

Reviewer #1 comments:

Comment-1:  The authors did not respond to my concerns seriously and appropriately. I would like to give them a second chance to revise the manuscript carefully.

Response: Thank you reviewer for your helpful suggestion. Authors have now modified the manuscript.

Reviewer 2 Report

1. The following information should be added in the manuscript:

" In earlier work (reference 4), we have used 0.2 M concentrations of both aniline and H2SO4. In the present study we have taken 0.1 M concentrations of both aniline and H2SO4. So, we get proper variations in the structure of polyaniline chain."

2. It seems that the statement below need some references. If the authors invented this method, it shoud be emphasized.

It is noteworthy that, the electro polymerization of aniline mostly involved into two stages. Firstly, a compact granular structure of PANI is formed on the SS electrode in the primary stage. Secondly, PANI further grows and then forms a uniform and loosely bound morphology in the advanced stage which seems to be spongy network like structure.

Author Response

Reviewer #2 comments:

Comment-1: The following information should be added in the manuscript: " In earlier work (reference 4), we have used 0.2 M concentrations of both aniline and H2SO4. In the present study we have taken 0.1 M concentrations of both aniline and H2SO4. So, we get proper variations in the structure of the polyaniline chain.

Response: Thank you reviewer for helpful suggestion. This sentence has been added in revised manuscript.

Comment-2: It seems that the statement below needs some references. If the authors invented this method, it should be emphasized. It is noteworthy that the electro polymerization of aniline mostly involved into two stages. Firstly, a compact granular structure of PANI is formed on the SS electrode in the primary stage. Secondly, PANI further grows and then forms a uniform and loosely bound morphology in the advanced stage which seems to be spongy network like structure.

Response: Thank you reviewer for helpful suggestion. The relevant reference has been cited in modified manuscript.

Round 3

Reviewer 1 Report

The authors still show no response to my concern. Based on their action, I refuse to further review this manuscript and provide any more comment. The decision making can be completed by the editor.